# Preference Parameters for the Calculation of Thermal Conductivity by Multiparticle Collision Dynamics

**DOI:** 10.3390/e23101325

**Published:** 2021-10-11

**Authors:** Ruijin Wang, Zhen Zhang, Long Li, Zefei Zhu

**Affiliations:** School of Mechanical Engineering, Hangzhou Dianzi University, Hangzhou 310018, China; zhangzhen@hdu.edu.cn (Z.Z.); lilong@hdu.edu.cn (L.L.)

**Keywords:** multiparticle collision dynamics (MPCD), coarse-grained, nanofluid, thermal conductivity (TC), parameterization investigation

## Abstract

Calculation of the thermal conductivity of nanofluids by molecular dynamics (MD) is very common. Regrettably, general MD can only be employed to simulate small systems due to the huge computation workload. Instead, the computation workload can be considerably reduced due to the coarse-grained fluid when multiparticle collision dynamics (MPCD) is employed. Hence, such a method can be utilized to simulate a larger system. However, the selection of relevant parameters of MPCD noticeably influences the calculation results. To this end, parameterization investigations for various bin sizes, number densities, time-steps, rotation angles and temperatures are carried out, and the influence of these parameters on the calculation of thermal conductivity are analyzed. Finally, the calculations of thermal conductivity for liquid argon, water and Cu-water nanofluid are performed, and the errors compared to the theoretical values are 3.4%, 1.5% and 1.2%, respectively. This proves that the method proposed in the present work for calculating the thermal conductivity of nanofluids is applicable.

## 1. Introduction

The main difficulty in heat transfer enhancement by nanofluids lies in the thermal conduction mechanism. Nowadays, not a few published results are contradictory and inconsistent because there are too many impact factors and many complicated underlying mechanisms. The microscopic molecular dynamics is the most common approach to calculate the thermal conductivity of nanofluids [1,2,3]. However, MD can be employed only in a small system due to the huge calculation workload. In our previous work, numerical simulations for calculating the thermal conductivity of Cu-Ar nanofluids by general MD were carried out. The computer’s running time for the case containing 6 Cu-nanoparticles with a size of 1.2 nm is more than a hundred hours [4,5]. This is far from enough to study the influence of particle aggregation on the thermal conductivity of nanofluids. In order to reduce the calculation workload, coarse-grained MD (CGMD) based on the Martini force field [6,7] was proposed and used in the simulations of characteristics of macromolecules such as sugar and amino acids. He et al. [8] employed the CGMD to calculate the viscosity of Cu-water nanofluid, and found that the calculation efficiency can be greatly elevated. However, the CGMD can still not be employed to calculate the transport coefficients of nanofluids in large systems. For example, it is still very hard to calculate the thermal conductivity of nanofluids containing aggregations of several hundreds of nanometers to several microns.

MPCD, sometimes referred to as stochastic rotation dynamics (SRD), was proposed by Malevanets and Kapral [9] in 1999. The computation workload in MPCD can be considerably reduced by coarsening the molecules of fluid, compared to the general MD. Moreover, MPCD can easily include thermal fluctuation and hydrodynamic interaction and be suitable to simulate the complex fluid, such as colloidal particles, polymers or electrolytes. De Angelis [10] verified that MPCD is a particle-based Navier-Stokes solver, and can be employed to simulate typical examples such as colloidal suspensions and polymer solutions. Nowadays, MPCD is extensively employed in many fields, such as soft matter [11,12], biological systems [13,14], colloidal suspensions [8,15,16] and so on. In addition, Dahirel [17] investigated the dynamical properties of solutes that are coupled to the fluid within the collision step, i.e., when local momentum exchange between fluid particles occurs. Batot [18] compared the transport coefficients of neutral and charged solutes in a model system by Brownian dynamics (BD) and SRD simulations. A hybrid method of MPCD-MD was proposed by Yamamoto [19] and employed to simulate the flow-induced structure of star polymers. Laganapan [20] employed SRD-MD to study the behavior of sheared colloidal suspensions with full hydrodynamic interactions. Du et al. [21] evaluated the influence of aggregation morphology on the thermal conductivity of nanofluid by the MPCD-MD hybrid method.

As for the dimensionless parameters of MPCD, they are generally set as: the mass of fluid m=1, the temperature ${\;k}_{B}T=1$, the bin size a=1, the rotation angle α=90–135°, the time-step h=1 and the average particle number in a cell (or number density) ρ=3–20 [12,13,14,22]. Sometimes, the parameters can map to the detailed simulation cases [19,23]. For example, in [19], the dimensionless mass m=1 can map to $1.44\times 10^{-10}$ g, $k_{B}T=1$ to $4.14\times 10^{-21}$ J, a=1 to 706 nm and so on. Such a mapping can result in deviation to transport coefficients such as viscosity and thermal conductivity. Hence, the Reynolds number, Mach number, Schmidt number and Peclet number should be verified before the simulations are conducted [24]. However, it is known from several publications [15,25,26,27,28] that the transport coefficients, such as the diffusion coefficient, the viscosity and the thermal conductivity generally vary with the parameters of MPCD, especially with the rotation angle, time-step and particle number density. Yamamoto [15] proposed a MPCD model to describe the effect of the colloidal particle volume fraction on the shear viscosity of suspensions for various MPCD parameters. Pooley [25] predicted the thermal conductivity in two and three dimensions, and found that greater deviation occurs when the rotation angle is near to 0 or 180° and the preferential value of number density is 3 for a shorter relaxation time. Ihle [26] showed how the Green-Kubo relations derived previously can be resumed to obtain exact expressions for the collision contributions to the transport coefficients. In addition, the collision contribution to the thermal conductivity, which becomes important for small mean free path and small average particle number per cell, is also derived. Kikuchi [27] showed that the viscosity has two contributions, streaming viscosity and collision viscosity. The former dominates at high temperatures and the latter at low temperatures. Lüsebrink [28] indicated that the prediction is better for systems with a large number density, large rotation angle and large time-step. For small values of rotation angle and time-step, the deviations decrease, probably due to a cancellation of errors. Moreover, the comparison among the three temperature gradient implementations is performed for the smaller number density.

It was found from the above review of the published results for simulating thermal conductivity of nanofluids by MPCD that the MPCD parameter selections have a great influence on the calculation of thermal conductivity [28,29]. The purpose of the present work is to calculate the thermal conductivity using MPCD so that a large system containing more particles can be simulated. Therefore, it is necessary to clarify the influence of MPCD parameters on computational results of thermal conductivity, and then the preferential values of MPCD parameters for water and argon can be determined. Finally, we attempt to extend the method to copper-water nanofluid.

## 2. Numerical Model

### 2.1. MPCD Implementation

In MPCD, the fluid consists of point-like particles, and the Navier-Stokes equation can be derived from the local mass and momentum conservation in the overall ensemble [10]. The mass, velocity and position of the $i^{th}$ point-particle are $m_{i},v_{i},\;r_{i}$, respectively. The update of particle positions and momenta can be defined in terms of successive streaming and collision step [30]. During the streaming step, the particles move ballistically in the absence of external forces, and the position update can be described by:(1)$$r_{i}(t+h)=r_{i}(t)+hv_{i}(t)$$
where the interval h is defined as time-step. In collision, relative velocities of all the particles are rotated by a given angle around a randomly chosen axis, so that their momentum can transfer inside a cubic bin (or sometimes named cell) with a size of a. The collision step is a simple non-physical scheme for ensuring momentum conservation. Multiparticle collisions within a bin are represented by the operation:(2)$$v_{i}^{*}=v_{i}+S_{\xi }D(\alpha )\left(v_{i}-{\bar{v}}_{i}\right)$$
where $v_{i}$ and $v_{i}^{*}$ are the velocities of the $i^{th}$ particle before and after collision, respectively. $S_{\xi }^{}$ and D(α) are operators to preserve temperature and to rotate randomly, and *α* is rotation angle. The stochastic rotation matrix, D(α), changes the magnitude and the direction of velocity of every particle to conserve the total mass, momentum and kinetic energy in the collision box [29], and there are several thermostats, $S_{\xi }^{}$, for temperature preservation, such as the Anderson thermostat [25,29,31]. In other words, this can ensure the presence of hydrodynamic interactions, together with thermal fluctuations [29,31]. Ripoll et al. [29] showed that α=130° in combination with a small time-step, h, leads to high Schmidt numbers, i.e., fluid-like behavior. It can be verified that this collision scheme conserves linear momentum and energy [28,30]. If rotation transformation is performed for all particles of solutes and solvents in a bin, the mean velocity in a bin reads as:(3)$${\bar{v}}_{i}=\frac{\sum_{i=1}^{N_{\xi }^{p}}m_{p}v_{i}+\sum_{i=1}^{N_{\xi }^{b}}m_{b}v_{i}}{m_{p}N_{\xi }^{p}+m_{b}N_{\xi }^{b}}$$
where $N_{\xi }^{p}$ and $N_{\xi }^{b}$ are the number of solutes and solvents in the $\xi^{th}$ bin, respectively. $m_{p}$ and $m_{b}^{}$ are the mass of solutes and solvents, respectively. Note that the Galilean invariance will be broken if the mean free path $\lambda =h\sqrt{k_{B}T/m}<a$, where ${\;k}_{B}\;$ is the Boltzmann constant, T is the temperature and m is the mass of fluid [30], which means that the particles repeat the collision in the same bin. Galilean invariance can be restored by a random shift of the cell grid before each collision step. In practice, the shift can be performed by moving all particles by a random vector whose components distribute uniformly in [a/2,−a/2]. However, this operation promotes the momentum transfer between the bins and results in larger transport coefficients [26]. Several collision rules are proposed by MPCD pioneers, such as MPC-SR–a, MPC-AT–a [32] and MPC-AT+a [33]. The angular momentum can be conserved in +a algorithms, rather than in –a algorithms. A collision rule that conserves both energy and angular momentum can be derived by Kikuchi [27].

### 2.2. Calculation of Thermal Conductivity

In equilibrium molecular dynamics, the thermal conductivity can be obtained by integrating the correlation function of the microscopic heat current based on the Green-Kubo formula [26]. Instead, in non-equilibrium molecular dynamics, the thermal conductivity can be obtained by Fourier’s law:(4)$$k=-\frac{Q}{\partial T/\partial x}$$
where Q is heat flux and ∂T/∂x is the temperature gradient in the x direction. Fast convergence in iterative computations can be expected in computation of thermal conductivity if the heat flux is calculated after the temperature gradient is imposed. In reverse, the convergence will be slowed down if the temperature gradient is calculated after the heat flux is imposed.

In MPCD, the local mass, momentum and energy can be defined, respectively, for each MPC bin as:(5)$$\rho =\frac{m}{a^{3}}\sum_{cell}1,\;j=\frac{m}{a^{3}}\sum_{cell}v,\;e=\frac{m}{a^{3}}\sum_{cell}\frac{v^{2}}{2}$$

However, the corresponding transport coefficients, such as viscosity, diffusion coefficient and thermal conductivity, can be obtained from the micro-scale transport during streaming and collisions. Hence, there exist both kinetic and collision contributions [34]. Two possible routes can be utilized to derive transport coefficients of the MPC fluid. One relates the transport coefficients to equilibrium fluctuations of the hydrodynamic fields. Another relates to nonequilibrium steady situations. In the present work, the “velocity exchange method” proposed by Muller-Plathe [35] was employed to calculate the thermal conductivity. The usual process to calculate thermal conductivity is: Firstly, divide the simulation box into a series of pieces with uniform thickness (Figure 1). The leftmost and rightmost ones are defined as the “heat sink” and the middle one as the “heat source”. After the hottest atom in the heat sink and the coldest one in the heat source are identified, these two atoms’ velocities are exchanged with each other [35]:(6)$$\left\{ \begin{array}{l} v_{1}^{*}=-v_{1}+2(m_{1}v_{1}+m_{2}v_{2})/(m_{1}+m_{2})\\ v_{2}^{*}=-v_{2}+2\left(m_{1}v_{1}+m_{2}v_{2}\right)/(m_{1}+m_{2}) \end{array}\right.$$
where the coldest particle, $m_{1}$, has the lowest velocity, $v_{1}$, and the hottest particle, $m_{2}$, has the highest velocity, $v_{2}$. Note that the relationship between velocity, v, and temperature, T, in microscopic reads as:(7)$$m\cdot v^{2}/2=3k_{B}\cdot T/2$$

Hence, a linear temperature profile or temperature gradient can be obtained after velocity exchange in a sufficient number of time-steps and reaching steady state. Besides, the calculation of heat flux can be implemented by integrating the energy exchange on the basis of velocity exchange as Equation (6). The sum of energy exchange or heat flux in a number of time-steps can be calculated by:(8)$$Q=\sum \frac{1}{2}\left[m_{1}\left(v_{1}^{2}-v_{1}^{*2}\right)+m_{2}\left(v_{2}^{2}-v_{2}^{*2}\right)\right]$$

Finally, the thermal conductivity of the nanofluid can be calculated by the resultant temperature gradient and heat flux based on the Equations (4) and (8).

### 2.3. Definition of Nondimensional Parameters

In the present simulation, the computations were carried out based on the rescaled variable according to the following rules: the position $x^{*}=x/a$, the time $t^{*}=t/\sqrt{m\sigma /\varepsilon }$, the mass of coarse-grained particle $M^{*}=M/m$, the temperature $T^{*}=k_{B}T/\varepsilon$ and the thermal conductivity $k^{*}=ka^{2}{/(k}_{B}T)$. Here, m is the mass of the argon molecule, σ and ε are the scale parameter and well depth respectively, of L-J potential for argon molecules and a is the bin size related to the size of the fluid atom (mapping to the argon atom in the present work). Note that the purpose of introducing σ and ε is to define nondimensional time and temperature, and they are set as σ=0.3405 nm and $\varepsilon =1.67\times 10^{-21}$ J. All simulation results mentioned below will be presented in these dimensionless units. The dynamic regime of the MPCD fluid depends on the input parameters, especially the rotation angle, α, the particle number density, ρ, and the time-step, h. In many published works, α=130°, 0.01≤h≤0.1 and number density ρ=10 are normally chosen because the fluid-like behavior can be presented in the MPCD framework [29].

## 3. Analysis of Various Effect Factors

It is known that the effect of MPCD parameters, such as mass of particle, bin size, time-step and rotation angle, on the computational results of thermal conductivity is notable and complicated. Hence, it is significant to determine the preferential values of MPCD parameters. In this section, the influence of various parameters on the calculation of thermal conductivity will be analyzed, then the preferential values of several key parameters such as time-step and rotation angle are determined for Ar, water and Cu-water nanofluid systems.

In the present work, the simulated box has a nondimensional size of 33.72 × 33.72 × 33.72, and is divided into 21 pieces (Figure 1). A microcanonical ensemble (NVE) is considered in the MPCD simulation. The system should be firstly brought to an equilibrium state after relaxation calculations in the canonical ensemble (NTV). A Nose-Hoover heating bath was employed during relaxation to maintain the temperature of all coarse-grained particles of the fluid. Then, the heat flux can be manually imposed. The data collections performed in NVE begin after equilibrium calculations to assure the system is in a steady state during the data collection [4]. All simulations were performed in the package of LAMMPS. In addition, in order to improve the precise calculation of thermal conductivity, the temperatures in heat source and heat sinks should not be considered because a great variation of temperature will be produced by the heat flux. The data collections for calculating the temperature gradient were performed in every piece every 10 time-steps. Then, an average temperature value can be obtained every 20 k time-steps by averaging the 100 values collected for every piece. Figure 2 shows the temperature distribution at 2000 k time-steps for the calculation of the temperature gradient.

### 3.1. Effect of Time-Step and Coarse-Grained Mass

A simulation box containing 62,500 coarse-grained particles (argon) was divided into 20 × 20 × 20 bins (Figure 2a): the bin size a=1.7, and the average number density ρ=9.11 (Figure 2b). All coarse-grained particles distribute in face-centered cubic (FCC) at the beginning of the MPCD simulations, so as to facilitate the control of average number density. For example, the lattice constant fcc=1.55 is for ρ=9.11 when a=1.7. The other parameters were temperature $T^{*}=0.71$ and rotation angle α=130°. Numerical simulations for various time-steps and grain masses were carried out to compute thermal conductivity. The time-step ranges 0.25~0.45 every 0.15, and the mass of grain ranges 1~2.5 every 0.5. The calculation method of thermal conductivity follows that in Section 2.2.

Figure 3 shows the temperature distribution in a simulation box at 2000 k time-steps and indicates that the symmetric temperature decreases linearly with the distance from the heat source in both right and left directions. Thus, the temperature gradient can be obtained. Figure 4 show us the data collections and calculations of thermal conductivity. It can be seen that the values of thermal conductivity for the right side, left side and their average fluctuate a lot at the beginning, and then the average values tend to stabilize after 1200 k time-steps. Therefore, we can obtain the thermal conductivity by averaging the last 100 values. The final value of thermal conductivity can be determined as 1.4742.

Figure 5 shows that the values of thermal conductivity vary with the time-step for various fluid coarse-grained particles. It can be seen that the values of thermal conductivity increase linearly with the time-step when $h^{*}=0.25–0.45$, no matter how large the fluid coarse-grained particle is. Moreover, Figure 5 also shows that the greater value of thermal conductivity can result in smaller coarse-grained particles when $M^{*}=0.4–10$. The most important result is that the slope of the fitting line of thermal conductivity also increases with the fluid coarse-grained particle. The slopes approach zero when $M^{*}>1.0$. The minimum slope (0.26551) means the preferential value at $M^{*}=4.0$ for argon.

### 3.2. Effect of Bin Size

The bin size, a, affects the calculation results of thermal conductivity considerably, because different bin sizes mean the average number of coarse-grained particles in each bin is different, even for the same system. The parameters such as bin size, lattice constant and number density are listed in Table 1. A larger lattice constant means a greater number density for an identical bin size. The other MPCD parameters for water are $M^{*}=4.0$ and $h^{*}=0.35$, respectively. Figure 6 shows that the thermal conductivity decreases with the increase of bin size for all lattice constants. This can be interpreted as follows: The energy exchange between coarse-grained particles in the same bin will increase if there are more coarse-grained particles due to the larger bin. Instead, the energy exchange between coarse-grained particles in different bins will obviously decrease, and this results in low thermal conductivity.

### 3.3. Effect of Rotation Angle

It can be found from [30] that the temperature jump approaches to zero and thermal conductivity tends to be stable when the rotation angle ranges between 120° and 180°. The rotation angles in almost all research works by MPCD were suggested to be 130° or 135°. However, no studies made an attempt to shed insight on the cases with a rotation angle larger than 180° or the cases with a combined rotation angle (CRA). Theoretically, a suitable CRA is beneficial to ensure that every particle is be involved in the collision, and consequently, to ensure a reasonable mean free path. Ten cases chosen for numerical simulations are listed in Table 2. The MPCD-related parameters for water were chosen as: the bin size a=1.25, the lattice constant fcc=1.55, the number density ρ=3.77, the temperature $T^{*}=0.71$, the mass $M^{*}=0.4$ and the time-step $h^{*}=0.35$. It can be seen from Table 2 that the average value of thermal conductivity is 3.1798, if only the cases without 360° are taken into account. The highest accuracy seen in cases 3 and 10 proves that the CRA for 90°, 180° and 270° with a probability of 1/3, 1/3 and 1/3 may be the preferential choice in rotation angle. In addition, the deviations are greater when the CRA includes 360°, because the coarse-grained particles do not take part in the collision. This signifies that the CRA including 360° is unreasonable. However, the results in the cases with 180° in CRA are surprisingly within reason.

### 3.4. Effect of Temperature

It is known that the thermal conductivity will increase with the temperature for most liquids. In order to verify that the same relationship can be obtained in the MPCD simulation, the calculations of thermal conductivity for a 33.72 × 33.72 × 33.72 system at temperature $T^{*}=0.5$, 0.71 and 1.0 were conducted. The MPCD-related parameters were set as: the mass of coarse-grained particle $M^{*}=1.0$, the time-step $h^{*}=0.35$, the bin size a=1.78, the combined rotation angle 90°, 180° and 270°, with 1/3 probability of each, and the lattice constant fcc = 1.25, 1.45, 1.55, 1.75 and 1.95. Figure 7 shows that the thermal conductivities at various temperatures vary with the lattice constants. It can be seen that the thermal conductivity increases with both the temperature and the lattice constant. These findings are consistent with the results in Section 3.3 and other published results. However, the thermal conductivity calculated by the MPCD simulation for various lattice constants will not always obey the above-mentioned rule. For instance, the thermal conductivity at $T^{*}=0.71$ and fcc=1.95 is greater than that at $T^{*}=1.0$ and fcc=1.55. This can be interpreted as follows: the greater kinetic energy of coarse-grained particles in a bin elevates the collision efficiency at a higher temperature if the number density of the simulation system is fixed. Nevertheless, more energy exchange can result at a lower temperature ($T^{*}=0.71$) than a higher temperature ($T^{*}=1.0$) if there are enough coarse- grained particles in a collision bin at fcc = 1.95. For most liquids, the higher the temperature, the greater the distance between the atoms or molecules, and the smaller the number density in the same bin size.

## 4. Thermal Conductivity Calculations

### 4.1. Thermal Conductivity of Ar

For the argon system, a simulation box of 33.72 × 33.72 × 33.72 containing 32,000 coarse-grained particles was simulated to determine the key MPCD-related parameters, such as rotation angle and time-step at fixed values: $M^{*}=1.0$, $T^{*}=0.71$, a=1.70 and fcc=1.55. The method and procedure to calculate thermal conductivity are the same as those described above. Four cases for various CRAs were simulated at various time-steps in the range of *h** = 0.7–1.1. CRAs of 90°, 180° and 270° have four combinations with a probability of (1/2, 1/2, 0), (1/3, 1/3, 1/3), (1/6, 2/6, 3/6) and (1/6, 1/6, 4/6). The calculation results are depicted in Figure 8. It can be seen that the thermal conductivity of liquid argon is 0.1365 W/(m·K) when the time-step $h^{*}=1.0$, and for CRAs 90°, 180° and 270° with a probability of (1/6, 1/6, 4/6). The deviation between the numerical result and the theoretical result (0.132 W/(m·K)) is 3.4%. For comparison, the results are converted back into ISO units on the basis of Section 2.2. Figure 9 shows the variation of thermal conductivity of liquid argon with the iteration time. Note that the thermal conductivity (0.1365 W/(m·K)) was obtained by averaging the last 100 values.

### 4.2. Thermal Conductivity of Water

Similar to liquid argon, the thermal conductivity of water was calculated in the same simulation box containing 23,328 coarse-grained particles by the same method and procedure. For convenient comparison, the parameters were fixed: $M^{*}=1.0$, $T^{*}=2.442$, a=1.70, fcc=1.55, $h^{*}=0.35$ and CRAs 90°, 180° and 270° with a probability of (1/6, 1/6, 4/6). The result of thermal conductivity is 0.6084 W/(m·K)), and the deviation from theoretical results (0.5990 W/(m·K)) is 1.5%.

### 4.3. Thermal Conductivity of Cu-Water Nanofluid

To investigate whether MPCD is suitable to calculate the thermal conductivity of nanofluids, a simulation box containing 14 Cu-nanoparticles was simulated (volume fraction 2.4 vol%). The parameters were fixed: $M^{*}=1.0$, $T^{*}=2.442$, a=1.70, fcc=1.55, $h^{*}=0.35$ and CRAs 90°, 180° and 270° with a probability of (1/6, 1/6, 4/6). However, the Green-Kubo formula was employed to evaluate the thermal conductivities of the nanofluid because the Muller-Plathe method assumes the system to be homogenous. The initial distribution of nanoparticles is shown in Figure 10a, and that after 2M time-steps is shown in Figure 10b. Figure 11 shows the variation of thermal conductivity of Cu-water nanofluid with the iteration time. It can be seen from Figure 5 that the thermal conductivity fluctuated wildly at the beginning, and then tended to stabilize. Hence, it is reasonable to perform the data collection in the thermal conductivity calculation in the last 1M time-steps. The thermal conductivity (0.6924 W/(m·K)) was obtained by averaging the last 500 values. The value of thermal conductivity by MD, 0.6839 W/(m·K) [5], is very close to that by MPCD, and the error is ~1.2%. Comparing to that of pure water, the thermal conductivity of Cu-water nanofluid improved by about 15.3%.

## 5. Conclusions

A MPCD method based on the velocity exchange method was proposed to calculate the thermal conductivity of water, argon and Cu-water nanofluid in the present work. Parameterization investigations on the calculation of thermal conductivity were performed to determine the preferential MPCD parameters. Subsequently, the calculations of thermal conductivity for liquid argon, water and copper-water nanofluid were conducted. The following conclusions can be drawn:(1)The method proposed is applicable as long as suitable MPCD parameters are selected. It is suitable for various systems, such as argon, water and nanofluids. The computational accuracy was ensured, and the deviations of argon, water and Cu-water nanofluid were 3.4%, 1.5% and 1.2%, respectively.(2)The combined rotation angle (90°, 180°, 270°) with a probability of (1/6, 1/6, 4/6) may be preferential for calculating the thermal conductivity. The reason could be the isotropy of the simulation system.(3)The adaptive time-steps were 1.0, 0.35 and 0.35 for argon, water and copper-water nanofluid respectively, because argon has a greater weight compared to water. The underlying mechanism may be the different interaction intensity between different particles. It can be interpreted by the different parameters, ε and σ, in L-J potential for different molecules.

## Figures and Tables

**Figure 1 entropy-23-01325-f001:**
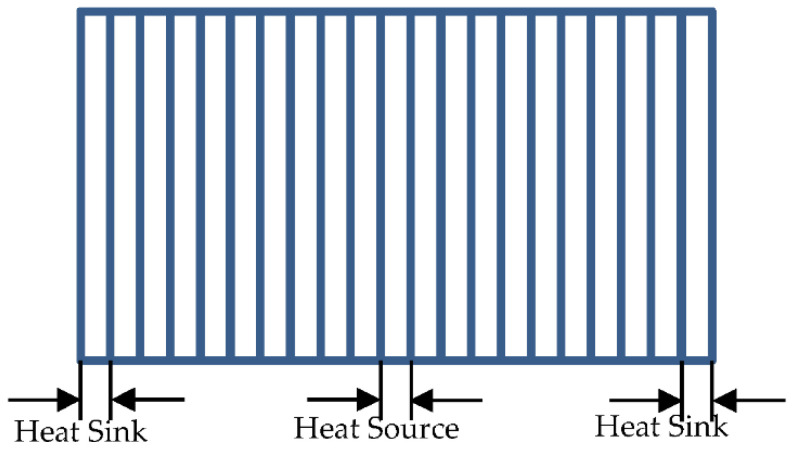
Divide the simulation box into 21 pieces with uniform thickness for calculating the temperature gradient. The middle one is heat source, and the leftest and rightest ones are heat sink.

**Figure 2 entropy-23-01325-f002:**
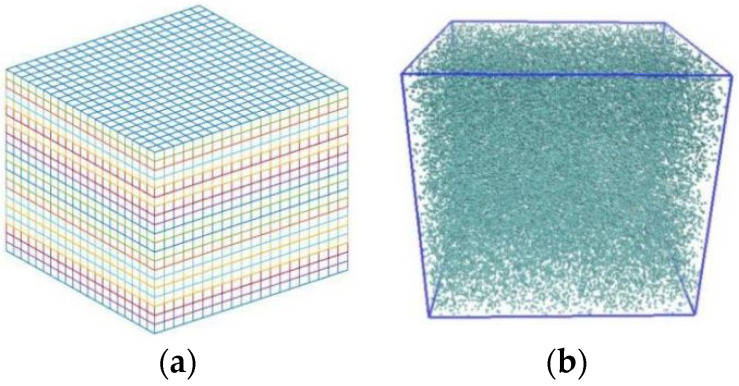
Schematic diagram of simulation system. (**a**) Divide the system into 8000 bins, and (**b**) 62,500 fluid coarse-grained particles in the simulation box.

**Figure 3 entropy-23-01325-f003:**
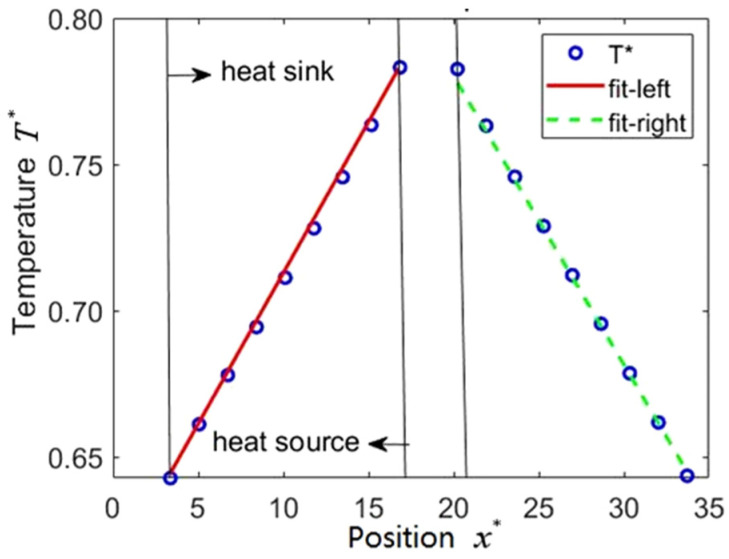
Temperature distribution between heat source and heat sink after 2000 k iterations (the fitting scheme is linear fitting, for the left one it can be written as *T** = 0.01020 *x** + 0.61307, and for the right one as *T** = −0.01018 *x** + 0.99921).

**Figure 4 entropy-23-01325-f004:**
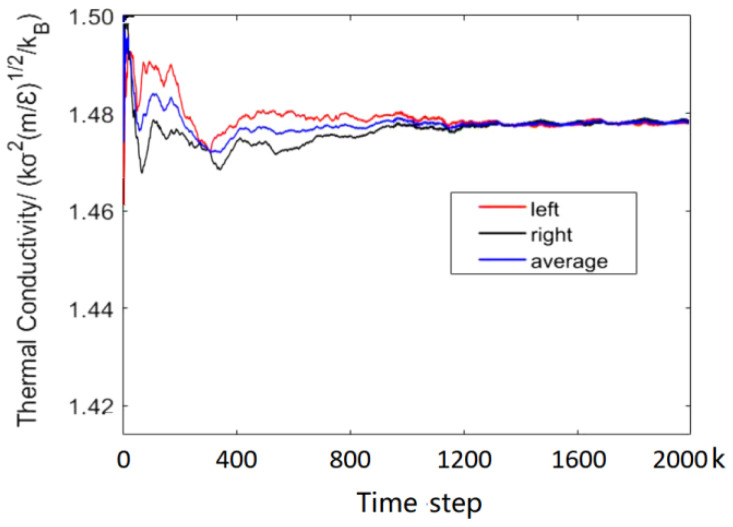
The variation of the thermal conductivity in the iteration process. There are two values of thermal conductivity, because there are two heat sinks. Take the average as the final value.

**Figure 5 entropy-23-01325-f005:**
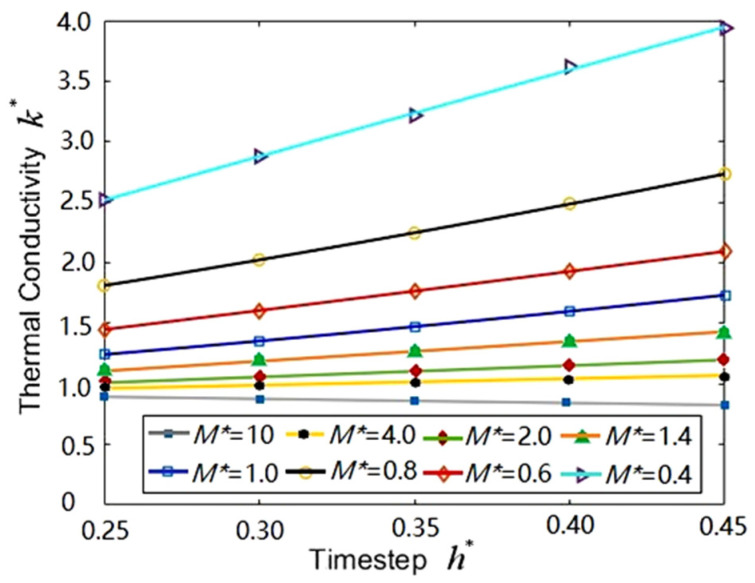
Thermal conductivities vary with the time-step for various fluid coarse-grained particles. The fitting scheme is a linear fit, and for *M** = 4, it can be written as *k** = 0.26551 *h** + 0.94038.

**Figure 6 entropy-23-01325-f006:**
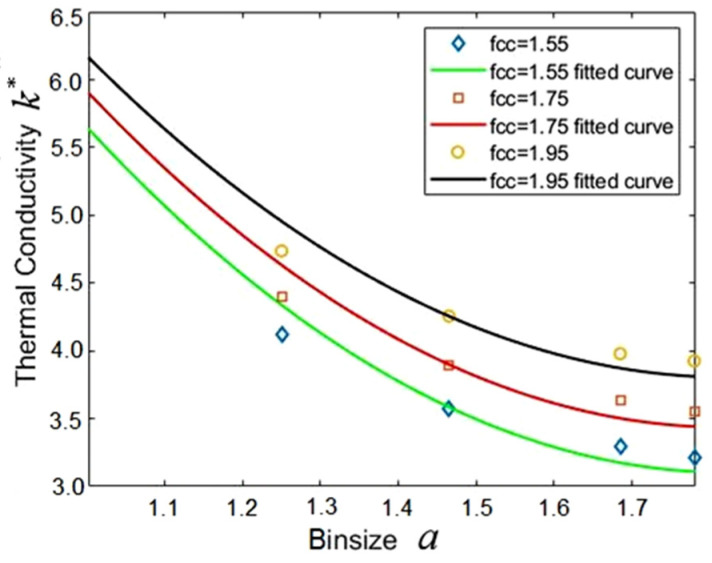
Thermal conductivity varies with the bin size for various lattice constants. The quadratic polynomial fitting: *k** = C_1_*a*^2^ + C_2_*a* + C_3_.

**Figure 7 entropy-23-01325-f007:**
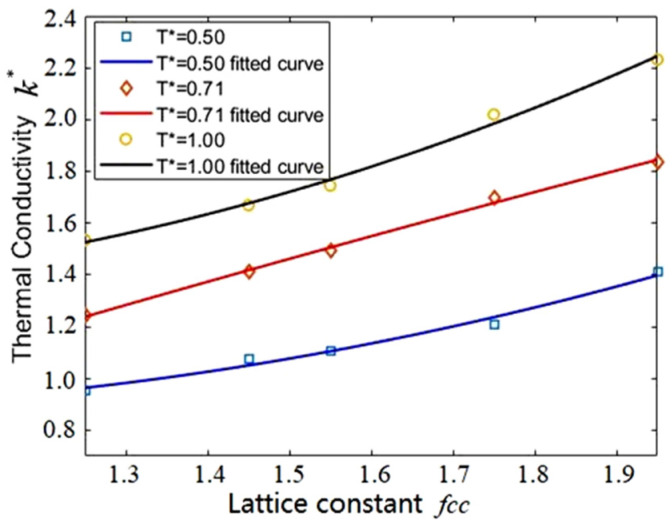
The thermal conductivity varies with the lattice constant for various temperatures. The quartic polynomial fitting: *k** = C_1_*fcc*^4^ + C_2_*fcc*^3^ + C_3_*fcc*^2^ + C_4_*fcc* + C_5_.

**Figure 8 entropy-23-01325-f008:**
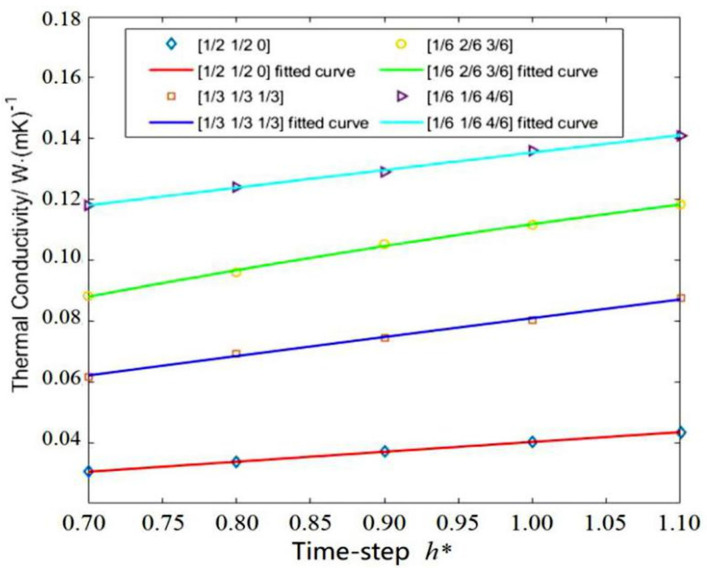
Thermal conductivity of liquid argon for various combined rotation angles and time-steps. The preferential value (0.1365 W/(m·K)) can be determined after comparing to the theoretical value (0.132 W/(m·K)). The fitting scheme is *k** = C_1_*h**^4^ + C_2_*h**^3^ + C_3_*h**^2^ + C_4_*h** + C_5_.

**Figure 9 entropy-23-01325-f009:**
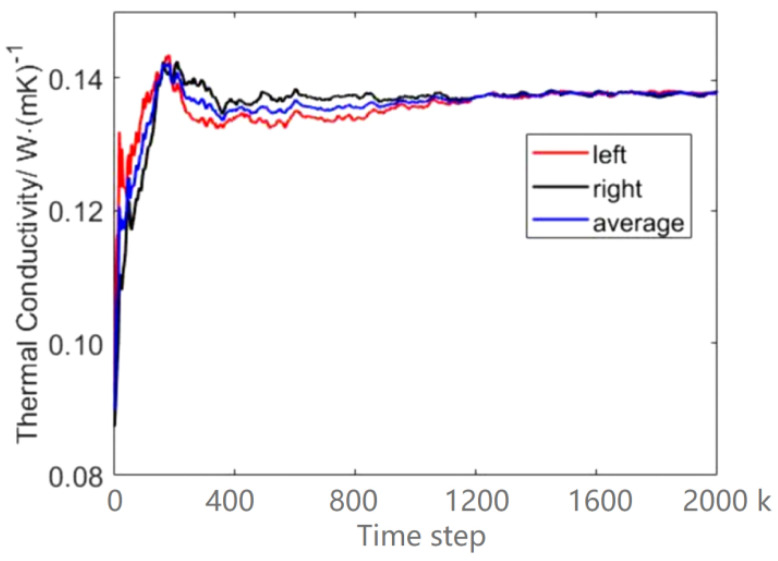
The fluctuations of the thermal conductivity in the iteration process. There are two values of thermal conductivity, because there are two heat sinks. Take the average as the final value.

**Figure 10 entropy-23-01325-f010:**
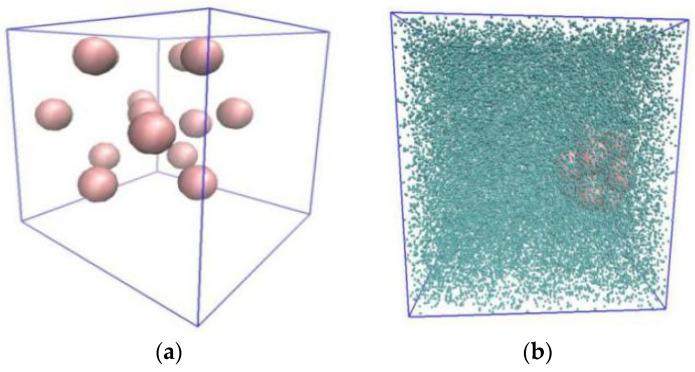
Initial distribution (**a**) and aggregation (**b**) of nanoparticles.

**Figure 11 entropy-23-01325-f011:**
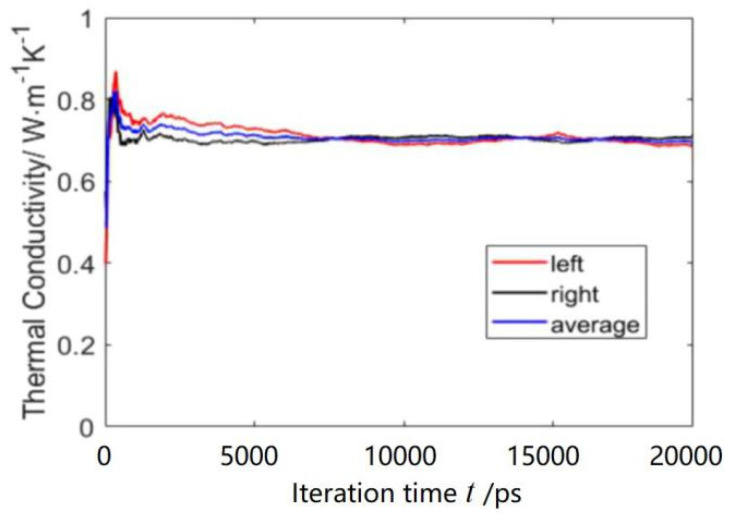
The calculation of thermal conductivity of Cu-water nanofluid.

**Table 1 entropy-23-01325-t001:** Number densities of water coarse-grained particles for various bin sizes and lattice constants.

**Lattice Constant**	***fcc* = 1.550**					
Bin size	1.000	1.250	1.465	1.685	1.780	1.983
Number density	1.590	3.370	5.140	7.810	9.110	12.720
**Lattice Constant**	***fcc* = 1.750**					
Bin size	1.000	1.250	1.465	1.685	1.780	1.983
Number density	1.782	3.778	5.771	8.788	10.250	14.310
**Lattice Constant**	***fcc* = 1.950**					
Bin size	1.000	1.250	1.465	1.685	1.780	1.983
Number density	2.003	4.000	6.470	9.840	11.500	16.020

**Table 2 entropy-23-01325-t002:** The results of thermal conductivity at various CRAs.

**Case Number**	**Case 1**	**Case 2**		**Case 3**			**Case 4**			**Case 5**		
CRA	130°	90°	180°	90°	180°	270°	90°	180°	270°	90°	180°	270°
Probability	1	1/2	1/2	1/3	1/3	1/3	1/6	2/6	3/6	1/6	1/6	4/6
TC	3.1241	3.1548		3.1476			3.2304			3.2714		
**Case Number**	**Case 6**	**Case 7**		**Case 8**			**Case 9**			**Case 10**		
CRA	135°	90°	270°	90°	270°	360°	90°	270°	360°	90°	270°	360°
Probability	1	1/2	1/2	1/3	1/3	1/3	1/6	2/6	3/6	1/6	1/6	4/6
TC	3.1453	3.2159		5.2538			7.0564			11.2008		

## Data Availability

Not applicable.

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
