# Peer review of "Preference Parameters for the Calculation of Thermal Conductivity by Multiparticle Collision Dynamics"

_entropy, 2021, doi:10.3390/e23101325_

Round 1
Reviewer 1 Report
The manuscript ‘Preferences Parameters for the Calculation of Thermal Conductivity in Nano-scale by Multiparticle Collision Dynamics’ presented using Multi-particle Collision Dynamics (MPCD) for the simulation of thermal conductivity in nanofluids, with the discussion on various effect factors including time step, binsize, rotation angle, and temperature. However, the current version of the manuscript is not suitable for publication. The detailed comments are as follows.
- The method lacks generality. The parameters (including ‘a’, ‘α’, ‘fcc’, ‘h’, etc) are not well justified, they are mostly randomly picked for the simulation of different materials. It would be better to have a trend or a range for selecting the values of the parameters to make the simulation more universal.
- Missing results. The authors mentioned the calculation of ‘Cu-water’ in the abstract and the intro paragraph in section 3, but there is no result of Cu-water presented in section 4.
- Figures. It would be better if the authors provided the fitting schemes of the data points in Figures 3 and 5-7.
- The manuscript needs extensive English editing as well as format editing. For example, the misuse of ‘references’ in the title; there are two section ‘4.1’s; the tense of the verbs are not consistent in the abstract (masses of grains and temperatures were carried out, and the influence of above parameters on thermal conductivity are analyzed), to name a few.
Author Response
To Reviewer1
The manuscript ‘Preferences Parameters for the Calculation of Thermal Conductivity in Nano-scale by Multiparticle Collision Dynamics’ presented using Multi-particle Collision Dynamics (MPCD) for the simulation of thermal conductivity in nanofluids, with the discussion on various effect factors including time step, binsize, rotation angle, and temperature. However, the current version of the manuscript is not suitable for publication. The detailed comments are as follows.
- The method lacks generality. The parameters (including ‘a’, ‘α’, ‘fcc’, ‘h’, etc) are not well justified, they are mostly randomly picked for the simulation of different materials. It would be better to have a trend or a range for selecting the values of the parameters to make the simulation more universal.
Response: Thanks for your constructive comments. In order to proposed a suitable method for evaluate the thermal conductivity of nanofluid containing more than 40 nanoparticles (we named it large system in manuscript), we have a great try to obtain preferential parameters such as ‘a’, ‘α’, ‘fcc’, ‘h’, etc. Like the preferential parameters of potential function used in MD will vary with the studied fluid, the MPCD parameters will be different also for different fluid(argon, water or nanofluid). It is very difficult to obtain a fixed value for different fluid. In revised manuscript, we added some interpretation on the election of parameters.
2)Missing results. The authors mentioned the calculation of ‘Cu-water’ in the abstract and the intro paragraph in section 3, but there is no result of Cu-water presented in section 4.
Response:Sorry, this is a stupid mistake. We deleted this content for “Cu-water” nanofluid due to the careless.
- It would be better if the authors provided the fitting schemes of the data points in Figures 3 and 5-7.
Response:Okay, we added some explanation on the data points.
- The manuscript needs extensive English editing as well as format editing. For example, the misuse of ‘references’ in the title; there are two section ‘4.1’s; the tense of the verbs are not consistent in the abstract (masses of grains and temperatures were carried out, and the influence of above parameters on thermal conductivity are analyzed), to name a few.
Response: Thanks. We check the manuscript carefully and revised.
Reviewer 2 Report
Dear Authors,
The content and the science in your work are really good. I feel you should rephrase your sentences in the Abstract and Conclusion sections. Y
Also, while using certain words like multi-particle or coarse-grained you should be consistent throughout your paper.
I do not understand the way you have cited the references in the introduction section. It does not give an impression of scientific writing.
Please change the title of your figures.
Overall, you really have good content but are not presented properly.
Thanks,

Author Response
To Reviewer 2
1)The content and the science in your work are really good. I feel you should rephrase your sentences in the Abstract and Conclusion sections.
Response: Thanks for your suggestions. We rewrote the abstract and conclusion sections.
2)Also, while using certain words like multi-particle or coarse-grained you should be consistent throughout your paper.
Response: Thanks for your suggestions. We revised.
3)I do not understand the way you have cited the references in the introduction section. It does not give an impression of scientific writing.
Response: Thanks for your critical comment. Our thought to prepare the introduction are, 1) The difficulty to calculate the thermal conductivity of nanofluid by MD due to the huge computation workload. Hence, a method to evaluate the thermal conductivity of nanofluid containing more nanoparticles or aggregation is required; 2) The MPCD method and its applications is introduced to show the availability of MPCD for calculating the thermal conductivity of nanofluid. 3) The selection of parameters of MPCD is normally solid for flow simulation. But, the selection of parameters of MPCD is influential to the calculation of the transport coefficients. 4) Our objectives.
We modified the introduction section in revised manuscript, especially for the third point.
4)Please change the title of your figures.
Response: Thanks for your suggestions. We revised.
- Overall, you really have good content but are not presented properly.
Response: Thanks for your positive comment.
Reviewer 3 Report
This work aimed an important point in nanoscale heat transfer however some general comments are as:
Nomenclature has to be added to show any parameters and abbreviations.
Conclusions section does not represent all the findings and it has to be extended.
The novelty of the work is not clear, and it has to be mentioned in the last paragraph of the introduction section.
Some quantitative results has to be mentioned in the introduction.
The main concern is the verification and the validation of the work with experimental results which has to be done. In the abstract validation ahs been mentioned but I could not find any experimental results in the paper.
Author Response
To Reviewer 3
This work aimed an important point in nanoscale heat transfer however some general comments are as:
- Nomenclature has to be added to show any parameters and abbreviations.
Response: Thanks for your suggestions. We added.
- Conclusions section does not represent all the findings and it has to be extended.
Response: Thanks for your constructive comments. We revised the conclusion section.
- The novelty of the work is not clear, and it has to be mentioned in the last paragraph of the introduction section.
Response: Thanks for your suggestion. We revised the introduction section.
4)Some quantitative results has to be mentioned in the introduction.
Response: Thanks. We revised the introduction section.
5)The main concern is the verification and the validation of the work with experimental results which has to be done. In the abstract validation has been mentioned but I could not find any experimental results in the paper.
Response: In order to verify the validation of our method, we compare our results with the theoretical results or MD results for argon, water and Cu-water nanofluid. The errors are 3.4%, 1.5% and 1.2%. In addition, the preferential values of MPCD parameter are also determined by comparing to the theoretical results. We added some interpretation and statement for verification in abstract and subsections 4.1,4.2 and 4.3.
Round 2
Reviewer 1 Report
The comments in the previous review are well addressed. The paper could be published in present form.
Author Response
Thanks!
Reviewer 3 Report
The comments have applied by authors.
Author Response
Thanks!
This manuscript is a resubmission of an earlier submission. The following is a list of the peer review reports and author responses from that submission.